# Regulation of Src Family Kinases during Colorectal Cancer Development and Its Clinical Implications

**DOI:** 10.3390/cancers12051339

**Published:** 2020-05-23

**Authors:** Wook Jin

**Affiliations:** Laboratory of Molecular Disease and Cell Regulation, Department of Biochemistry, School of Medicine, Gachon University, Incheon 406-840, Korea; jinwo@gachon.ac.kr

**Keywords:** Src family kinases, colorectal cancer, metastasis, epithelial–mesenchymal transition, cancer stem cells, drug targeting

## Abstract

Src family kinases (SFKs) are non-receptor kinases that play a critical role in the pathogenesis of colorectal cancer (CRC). The expression and activity of SFKs are upregulated in patients with CRC. Activation of SFKs promotes CRC cell proliferation, metastases to other organs and chemoresistance, as well as the formation of cancer stem cells (CSCs). The enhanced expression level of Src is associated with decreased survival in patients with CRC. Src-mediated regulation of CRC progression involves various membrane receptors, modulators, and suppressors, which regulate Src activation and its downstream targets through various mechanisms. This review provides an overview of the current understanding of the correlations between Src and CRC progression, with a special focus on cancer cell proliferation, invasion, metastasis and chemoresistance, and formation of CSCs. Additionally, this review discusses preclinical and clinical strategies to improve the therapeutic efficacy of drugs targeting Src for treating patients with CRC.

## 1. Introduction

The purification of v-Src (pp60src) from Rous sarcoma virus (RSV), which activates the insulin receptor, led to the identification of various Src family kinases (SFKs) [1,2], such as Src, Fyn, Yes, Yrk, Fgr, Hck, Blk, lymphocyte-specific protein tyrosine kinase (Lck), and Lyn. SFKs are a group of non-receptor kinases. Src comprises a unique amino-terminal domain, a carboxy-terminal negative tyrosine regulatory region (Tyr530), and four Src homology (SH) domains, namely SH4, SH3, SH2, and SH1 (kinase) domains. Src is activated upon autophosphorylation of the SH1 domain at Tyr419. Src is inactivated by the interaction between the SH2 domain and phosphorylated Tyr530 [2,3,4].

Src is involved in the pathogenesis of colorectal cancer (CRC). Src was first identified in patients with CRC. The average Src expression levels in CRC patients are reported to be 2.4-fold higher than those in the normal mucosa. Additionally, the levels of Src in liver metastases of CRC are increased by 2.2-fold compared to those in the normal mucosa. Furthermore, the expression levels of Src in extrahepatic colorectal metastases are markedly upregulated (12.7-fold) when compared to those in the normal mucosa [5,6]. The expression of Src was reported to be upregulated in 95% of patients with adenomatous mucosa, 32% of patients with synchronous metastatic lesions, and 14% of patients with adjacent carcinomatous mucosa [7]. One study reported that moderate and high expression levels of Src are observed in 21% and 9% of patients with CRC, respectively, and that enhanced Src expression is closely associated with poor survival [8]. Moreover, Src upregulation markedly promotes tumor growth in vivo. The highly metastatic CRC cells exhibit markedly upregulated Src expression when compared to the poorly metastatic CRC cells [9]. 

The expression levels of c-Yes and Src were reported to be upregulated in 64% and 95% of primary patients with CRC, respectively. Additionally, the expression levels of Src and c-Yes were reported to be markedly upregulated in 71% and 10% of liver metastases of CRC, respectively. Patients with CRC exhibiting high c-Yes expression are associated with a worse prognosis than those exhibiting low c-Yes expression [10]. Moreover, c-Yes promotes the migration of CRC cells without affecting the proliferation of CRC cells [11].

Lck, a member of the SFK family, plays a vital role in the activation and maturation of developing T-cells [12]. In addition to the CRC cells, the expression of Lck was reported to be upregulated in 78% of patients with CRC [13,14].

Several biological processes, including upregulation and activation of SFKs, are essential for the progression of CRC. These processes confer the CRC cells with several characteristics, including tumor-forming and metastatic abilities and resistance to therapeutic agents. This review summarizes recent and new advances in the understanding of Src-mediated CRC progression. Additionally, this review discusses new challenges in anti-cancer therapies for targeting Src activation during CRC progression.

## 2. Receptor-Mediated Signaling Pathways Activate Src during the Progression of CRC

Various membrane receptors and their ligands play a central role in activating Src through different mechanisms. The activation and upregulated expression of the receptors, and the resulting activation of Src, promote proliferation, invasiveness, and malignancy of CRC cells.

### 2.1. Epidermal Growth Factor Receptor (EGFR)

Epidermal growth factor (EGF) and hepatocyte growth factor (HGF) activate Src through the formation of Src/EGFR and c-Met receptor complexes, respectively, which markedly enhance the invasiveness of CRC cells (Figure 1). The activated Src upregulates the expression of c-Met [15,16,17], while EGF promotes the interaction between human epidermal growth factor receptor 2 (HER2) and Src in the CRC cells [15]. EGF-mediated Src activation inhibits caspase-8 activity by Fas-induced Tyr380 phosphorylation of caspase-8, which suppresses apoptosis of CRC cells. The expression levels of Src and Tyr380-phosphorylated caspase-8 were reported to be upregulated in 80% and 82.6% of patients with CRC, respectively, [18]. Additionally, the activation of Src and EGFR by Erb-B2 Receptor Tyrosine Kinase (ERBB4), which was reported to be upregulated in 43% of patients with CRC, results in the upregulation of cyclooxygenase 2 (COX2) expression, which enhances the proliferation of CRC cells [19,20]. Moreover, activation of the EGF/EGFR/Src/NADPH oxidase signaling axis results in phosphorylation of NF-κB by AKT, which induces the translocation of NF-κB from the cytosol to the nucleus. Further, activated NF-κB upregulates the expression of heme oxygenase-1 (HO-1) by binding to the promoter of *HO-1* genes, which suppresses the apoptosis of CRC cells [21]. EGF-induced Src activation enhances the localization of pseudopodium-enriched atypical kinase 1 (PEAK1), which was reported to be markedly upregulated in 81% of patients with CRC, to the actin cytoskeleton and focal adhesion. PEAK1 promotes cell proliferation, migration, and tumor growth by activating paxillin (PXN), p130Cas and ERK [22].

Epidermal growth factor receptor kinase substrate 8 (Eps8), which is an adaptor protein of tyrosine kinase receptors, including EGFR, is reported to be involved in the pathogenesis of cancer [23]. The expression of Eps8 was reported to be upregulated in 62% of patients with CRC. Additionally, advanced stages of CRC are associated with markedly higher Eps8 expression levels than the early stages of CRC. Furthermore, the expression of Eps8 is correlated with that of Src and focal adhesion kinase (FAK). Eps8 induces the proliferation and growth of CRC cells by promoting the formation of the Eps8/Src complex, which activates FAK. The proliferation of CRC cells is also regulated by Eps8-mediated activation of the signal transducer and activator of transcription 3 (STAT3) and mTOR, which upregulate the expression of FAK [24].

### 2.2. Vascular Endothelial Growth Factor Receptor (VEGFR) and Fibroblast Growth Factor Receptor (FGFR)

The upregulation of vascular endothelial growth factor (VEGF) and Src expression is essential for vascularization of CRC (Figure 1). The loss of Src downregulates VEGF expression and subsequently suppresses vascularization of CRC [25]. VEGF promotes the activation of SFKs, including Src and Yes, by promoting the formation of the VEGFR-1/SFK complex. This complex promotes the migration of CRC cells through the activation of downstream targets, including FAK, p130^cas^, and PXN. However, treatment with IMC-18F1, a VEGFR-1 inhibitor, suppresses the migration of CRC cells without affecting cell proliferation [26]. Moreover, the upregulation of VEGF by Src-mediated K-Ras activation, under hypoxic conditions, enhances vascularization and cell proliferation of CRC [27].

FGFR4 is involved in the Src-mediated pathogenesis of CRC. Knockdown of FGFR4 or treatment with TKI258, an FGFR inhibitor, markedly inhibits Src activation, which results in the loss of metastatic potential, epithelial–mesenchymal transition (EMT) induction, and tumor growth in vivo [28,29].

### 2.3. Interleukin (IL)-4/IL-13/IL-13Rα2

Expression levels of IL-13 receptor (IL-13Rα2) and its ligands (IL-4 and IL-13) are upregulated in patients with CRC, which are closely correlated to advanced tumor stages and poor survival. IL-13Rα2 interacts with a family with sequence similarity 120A (FAM120A), and forms protein network associations with FAK, Src, PI3K, G-protein-coupled receptors (GPCRs) and the TNFRSF10B (DR3) receptor. These interactions activate the Src and PI3K/AKT pathways, which promote the invasion and liver metastases of CRC cells in vivo. Additionally, FAM120A enhances liver metastases and viability of CRC cells in vivo (Figure 1) [30,31].

### 2.4. IL-6 Signal Transducer (IL6ST), IL-6R and IL-11R

Adenomatous polyposis coli (APC), a negative regulator of the Wnt/β-catenin signaling pathway, functions as a tumor suppressor by directly interacting with β-catenin and decreasing its stability [32]. *Apc^−/−^* mouse small intestinal (SI) organoids and CRC cells exhibit upregulated expression levels of IL6ST, IL-6R and IL-11R. The positive autoregulatory loop of APC markedly promotes the activation of Src, YAP, Notch, and STAT3 (Figure 1). The treatment combination of the SFK inhibitor (PP2 and AZD0530) and Janus-activated kinase (JAK) inhibitor (ruxolitinib) effectively decreases the tumor number and size, as well as the growth of CRC cells [33]. Regorafenib is a sorafenib-derived chemotherapy drug as a multikinase inhibitor, which blocks the activity of (VEGFR1–3), PDGFRβ, BRAF, KIT, RET, the angiopoietin 1 receptor (TIE2) and FGFR1 [34]. The treatment combination of ruxolitinib, regorafenib and VEGFR2-TIE2 inhibitors does not improve the overall survival (OS) or progression-free survival (PFS) of patients with relapsed and metastatic CRC (phase II study) (ClinicalTrials.gov Identifier: NCT02119676) [35].

### 2.5. G-Proteins/GPCR

G-proteins and GPCRs regulate cell survival through Src-mediated activation of multiple signaling pathways (Figure 1). Prostaglandin E2 (PGE2), a bioactive lipid molecule produced by cyclooxygenase enzymes (COX-1 and COX-2), promotes the activation of EGFR through matrix metalloproteinase (MMP)-2 and MMP-9 activation, which results in the activation of Src/MAPK/ERK signaling pathway [36]. The interaction between PGE2 and EP1, a GPCR, activates AKT, which promotes the activation of ERK and Src and upregulates the expression of VEGF by inducing the expression of hypoxia-inducible factor 1α (HIF-1α) [37]. PEG2-mediated activation of EGFR/AKT induces COX-2 expression by enhancing the binding of Sp1 to the −245/−240 COX-2 promoter, which activates the EGFR/p38 MAPK and Src-ERK1/2-Sp1 signaling pathways [38,39,40].

Protease-activated receptors (PARs), which are GPCRs activated by proteases, contribute to the progression of cancer through extracellular matrix degradation [41]. PAR-1-mediated activation of Gαo/Gαi, the pertussis toxin (PTx)-sensitive G-proteins, suppresses the Src and c-MET-mediated enhanced invasiveness of CRC cells [42]. However, the formation of the PAR-1/Gα12/Gα13 complex promotes cell invasion through the activation of Src, which is mediated by the activation of the RhoA/Rho-associated kinase (ROCK)-dependent signaling pathway [43]. The activation of PAR-1 and PAR-2, by thrombin and the serine protease trypsin or the specific PAR2-activating peptide (AP2), respectively, results in the activation of EGFR through MMP-dependent cleavage and release of transforming growth factor (TGF)-α, which promotes cell proliferation by activating ERK1/2 [44,45]. Moreover, the activation of PAR4, whose expression is reported to be upregulated in 76% of patients with CRC, by Src promotes cell proliferation through ERK1/2, by activating HER2 [46].

CCK_2i4sv_R, a GPCR, is a splice variant of the gastrin receptor (CCK_2_R). CCK_2_R, which is reported to be involved in the pathogenesis of human colon and pancreatic cancers [47], activates Src. In contrast to CCK_2_R, the third intracellular loop (3il) domain of CCK_2i4sv_R activates Src and FAK by forming a complex with Src [48]. Additionally, the activation of CCK_2_R, by ligands such as glycine-extended gastrin (G-Gly) and progastrin, promotes cell proliferation by promoting the Src-mediated activation of ERK, PI3K/AKT, and JAK 2/STAT3, and upregulating the expression of tumor necrosis factor (TNF)-α [49,50,51].

Src-induced Rac1 activation results in the generation of reactive oxygen species (ROS), which are involved in various cellular receptor signal transduction pathways. Src increases the phosphorylation of tyrosine in Vav2 and Tiam1, which are Rac1 guanine nucleotide exchange factors. Vav2 and Tiam1 activate Rac1, a monomeric G-protein, which subsequently activates multiple signaling pathways that regulate gene expression and promote cell transformation [52]. Wnt3α induces the formation of Src/axin-binding protein Dishevelled (Dvl) complex to activates Src and then induces NADPH oxidase (Nox)-1-mediated ROS generation through Vav2-mediated Rac1 activation. Increased ROS production inactivates nucleoredoxin (NRX) by enhancing NRX oxidation, which inhibits NRX-Dvl complex-mediated β-catenin degradation, and consequently activates the Wnt-β-catenin signaling pathway [53,54]. Src-mediated ROS generation promotes the Nox1-mediated surface expression of α_2_β_1_ integrin, which increases the migration of CRC cells [55].

## 3. Modulators That Activate Src during the Progression of CRC

The activation of Src can be induced by components of intracellular signaling pathways. Additionally, oncoproteins, which are activated during CRC progression, modulate Src.

### 3.1. Aryl Hydrocarbon Receptor (AhR) Signaling

AhR plays an essential role in the Src-mediated tumorigenesis of CRC (Figure 2A). AhR is activated by the ligand TCDD. Activated AhR functions as a transcriptional factor to promote the expression of Src by direct interaction. Subsequently, this promotes the proliferation of CRC cells by activating COX2 and ERK1/2 through Src-mediated EGFR activation [56]. Additionally, Src-mediated ERK1/2 activation suppresses apoptosis of CRC cells through ubiquitin-mediated degradation of Bik, a death accelerator [57].

### 3.2. Receptor-Type Protein Tyrosine Phosphatase (PTPR) Positively Regulates Src

The upregulation of protein tyrosine phosphatase (PTP) contributes to the pathogenesis of CRC (Figure 2A). PTP1B and PTP1B truncation mutants enhance Src activity. CRC cells exhibit marked upregulation of PTP1B. PTP1B-induced Fyn C-terminal peptide (FCP) phosphatase activity enhances Src activity by inhibiting the phosphorylation of Tyr530 in Src. PTP1B inhibitors decrease the metastatic potential and tumor growth by inhibiting Src activation [58]. Additionally, six PTP1B truncation mutants were identified in 2–21% of patients with CRC. The PTP1BΔE6 mutant, which lacks the phosphatase activity, induces the Src-independent transformation of rat embryo fibroblasts (REF) [59].

Receptor PTPα (RPTPα), which is required for the Src-mediated survival of CRC cells, activates Src. RPTPα splice variants (RPTPα245 and 445), which lack the D2 domain and contain additional spurious residues, were detected in patients with CRC. RPTPα245-mediated dephosphorylation and activation of Src transformed the REF in vitro and in vivo [60].

Phosphatase of regenerating liver 3 (PRL-3), a tyrosine phosphatase, is reported to be involved in the pathogenesis of CRC. CRC cells were reported to exhibit upregulated expression of PRL-3 when compared to non-metastatic CRC cells or normal colon epithelium [61]. PRL-3 and CD31 are endothelial markers. The expression of PRL-3 in CRC cells promotes angiogenesis in vivo and in vitro by recruiting human umbilical vascular endothelial cells (HUVECs) [62]. The loss of PRL-3 inhibits VEGF-mediated Src activation and decreases the vascular cell invasiveness, which affects the tumor vasculature in a CRC mouse model [63]. Moreover, the phosphorylation of Tyr53 in PRL-3 by platelet-derived growth factor (PDGF)-activated Src is essential for the activated RhoC-mediated invasion of CRC [64].

### 3.3. Other Modulators of Src

Methionine aminopeptidase 2 (MetAP2), a metalloprotease, promotes myristoylation of oncoprotein and angiogenesis. MetAP2 is reported to be a therapeutic target for cancer and obesity [65,66]. One study reported that the expression of MetAP2 was upregulated in 100% of patients with CRC [67], whereas another study reported that MetAP2 expression was upregulated in 66% [68] of patients with CRC. Treatment with the NC2213, a MetAP2 inhibitor, markedly inhibited the activation of Src, a target of MetAP2 in CRC cells (Figure 2A) [68,69]. The role of MetAP2 in the progression of CRC is not known. However, the therapeutic efficacy of M8891, a MetAP2 inhibitor, was evaluated for CRC (ClinicalTrials.gov identifier: NCT03138538).

Aberrant glycosylation is a common feature of cancer cells. In mammalian cells, four types of sialidases have been identified. Ganglioside is modulated by plasma membrane-associated sialidase (Neu3), which is a glucosidase involved in the suppression of apoptosis of CRC cells. One study reported that the expression and activity of Neu3 were upregulated in all 32 patients with CRC. The activation of EGFR downstream targets, including Src, ERK and AKT pathways, by Neu3, induces the expression of Bcl-2 and suppresses the expression of caspase, which promotes cell survival (Figure 2A) [70,71,72]. Compared to Neu3, Neu4 exerts opposing effects in CRC. In patients with CRC, the average expression levels of Neu4 in the normal mucosa are 2.8-fold higher than those in the tumors. The upregulated expression of Neu4 induces apoptosis of CRC cells. EGF-induced activation of Src suppresses Neu4 expression by activated Wwox-mediated inhibition of p73 and AP2γ activities [73].

Resistance against 5-fluorouracil (5-FU) is associated with the upregulated expression of heat shock protein (HSP)90. The upregulation of HSP90 activates Src and upregulates the Src downstream target thymidylate synthase (TYMS), which subsequently increases tumor growth in vivo (Figure 2A). However, treatment with 17-AAG, an HSP90 inhibitor, and dasatinib, an Src inhibitor, markedly decreased the growth and lung metastasis of 5-FU-resistant CRC tumors in vivo [74].

Diacylglycerol kinase (DGK) α, which plays a crucial role in the signaling and metabolic pathways, regulates the level of two lipids through diacylglycerol-mediated production of phosphatidic acid. DGKα induces tumor growth in vivo through the formation of a DGKα/Src complex, which activates Src. However, the treatment combination of R59949, a DGK inhibitor, and PP2, an Src inhibitor, markedly enhances apoptosis when compared to the individual treatments (Figure 2A) [75].

Poliovirus receptor (PVR, CD155) is reported to be a therapeutic target for tumors. CD155, which promotes metastatic potential, is associated with poor prognosis in various types of cancer [76,77]. One study reported that the expression of CD155 is upregulated in 86.6% of patients with CRC, and that CD155 induces proliferation, migration, and invasion of CRC cells in vivo and in vitro by upregulating the expression of MMP2, FAK and Src (Figure 2A) [78].

## 4. Src Activates Src Substrates during the Progression of CRC

Several studies have demonstrated that the Src-mediated upregulation of various downstream proteins, and Src-mediated activation of intracellular signaling pathways, are involved in the progression of CRC (Figure 2A). The phosphoproteomic analysis revealed that protein tyrosine receptor kinases (Met, EphA2 and EphB2), non-receptor tyrosine kinases (FAK and Syk) and pseudo-tyrosine kinases (SgK223 and SgK269) are the targets of Src. Src activates Met, EphA2, FAK, Syk, and SgK223 by forming a complex. These proteins increase the invasion of cells [79].

The activation of Src has been shown to be markedly correlated with urokinase receptor (u-PAR) expression and decreased survival of patients with CRC. The enhanced expression of u-PAR is associated with low survival of patients with CRC [80]. The activation of Src promotes the transcriptional activation of u-PAR, through Src-induced binding of Sp1 to the promoter region −152/−135 of u-PAR, which enhances u-PAR expression and facilitates u-PAR-mediated degradation of the extracellular matrix [81]. Moreover, activated Src activates c-Jun N-terminal kinase (JNK), which subsequently activates c-Jun and upregulates the transcription of u-PAR by enhancing the binding of AP1, phospho-c-Jun and Fra-1 to the −190/−170 region (AP-1 motif) of the u-PAR promoter. The activation of Src is markedly correlated with AP-1 binding and u-PAR expression in patients with CRC [82]. The downregulation of u-PAR expression does not affect cell proliferation. However, the downregulation of u-PAR expression suppresses the progression of CRC, by inhibiting the activation of Src and ERK/MAP kinase, the secretion of u-PA and MMP-9, and the plasminogen-dependent extra matrix proteolysis [83].

PXN promotes the progression of CRC. One study reported that the expression of PXN is upregulated in 65% of patients with CRC, and that PXN expression is closely correlated with the recurrence of CRC [84]. Activated PXN promotes the Src-mediated phosphorylation of Tyr31/Tyr118 in PXN, which increases the migration of CRC cells. Src-mediated PXN activation results in the activation of p21 protein-activated kinase 1 (PAK1). PAK1-mediated phosphorylation of Ser272 in PXN is essential for the interaction between PXN and Bcl-2, and the stabilization of Bcl-2 against ubiquitin-mediated degradation. The treatment combination of 5-FU and IPA-3, a PAK1 inhibitor, strongly inhibits tumor formation by 5-FU-resistant CRC cells [85]. Src-mediated activation of PXN results in the activation of FAK, which enhances extracellular pressure and promotes cell adhesion to type I collagen [86]. Src-mediated activation and translocation of PI3K lead to enhanced cell adhesion by promoting the formation of FAK, a regulatory (p85) subunit of PI3K/integrin β1 [87]. 

The expression of T-LAK cell-originated protein kinase (TOPK), which is reported to be involved in the progression of cancer, is upregulated in various cancers, including CRC, and is associated with poor clinical outcomes [88]. EGF-induced Src activation results in the phosphorylation of Tyr74 in TOPK by direct interaction. Src activates TOPK by promoting TOPK stability. Src-induced TOPK stability promotes tumor growth in vivo and in vitro [89].

Calcium-independent protein kinase Cδ (PKCδ) functions as a tumor suppressor. The overexpression of Src results in the downregulation of PKCδ, which promotes tumor transformation [90].

The expression of ADAM28, a metalloprotease, and phospho-Src is reported to be upregulated in 50% of patients with CRC. The v-Src-activated PI3K/AKT and ERK1/2 signaling pathways induce ADAM28 expression, which is correlated with the survival and progression of cancer [91].

## 5. Tumor Suppressors Inhibit Src Activation during the Progression of CRC

Several studies have reported the role of multiple intracellular regulators in inhibiting proliferation and metastasis of CRC cells, by blocking the activation of Src (Figure 2B).

### 5.1. C-Terminal Src Kinase (CSK)

CSK is a negative regulator of Src. The activation of Src is inversely correlated with the activity of CSK in patients with CRC (correlation coefficient = −0.71) [92]. One of the early events associated with the pathogenesis of CRC is the downregulation of CSK expression, which markedly reduces the formation of preneoplastic colonic mucosa. The inhibition of CSK enhances the Src-mediated activation of the Raf/MAPK signaling pathway, which promotes the proliferation of CRC cells [93]. CSK also inhibits the activation of Lyn and Fyn by promoting Tyr508 phosphorylation in Fyn and Lyn [94].

Membrane localization of CSK is essential for the regulation of Src activity. CRC cells exhibit markedly decreased CSK membrane localization and enhanced cytoplasmic localization. Phosphoproteins associated with the glycosphingolipid-enriched microdomain (*PAG*)-induced membrane localization of CSK markedly inhibit the invasion of CRC cells [95]. One study demonstrated that upregulated CSK expression reduces the invasion and lung metastases of CRC cells via MMP-2 by inhibiting the activation of Src [96]. The overexpression of CSK negative mutant in the CRC cells induces a mesenchymal cell-like phenotype, cell proliferation and metastatic ability, through the activation of SFK, which activates the integrin-SFK-mediated cell adhesion signaling [97].

The activation of p140Cap, an Src adaptor protein, results in the upregulation of CSK activity and decreased cell migration through the formation of the Src/CSK complex, which inhibits Src activity [98]. The interaction between EGFR and E-cadherin increases in the presence of p140Cap, which inhibits the activation of EGFR, Src and Ras. Subsequently, this inhibits the EGF-mediated activation of ERK1/2, which inhibits the proliferation and migration of CRC cells [99].

### 5.2. PTPRs

Protein tyrosine phosphatases (PTPRs) negatively regulate the progression of CRC. The expression of protein tyrosine phosphatase receptor type O (PTPRO) was upregulated in patients with CRC who exhibited a partial response to cetuximab and stable disease, when compared to that in patients exhibiting progressive disease. The formation of the PTPRO/EGFR complex inhibits Tyr845 phosphorylation in EGFR by directly dephosphorylating Src at Tyr416, which inhibits the progression and drug resistance of CRC cells [100].

The knockout of PTPRT dephosphorylates PXN at Tyr38, which results in a marked increase in tumor size in the APC^min+/−^ mice. However, Src-induced phosphorylation of PXN at Tyr38 increases the dasatinib resistance of tumor cells by activating the p130cas/PI3K/AKT signaling pathway [101].

### 5.3. miRNAs Negatively Regulate Src

MicroRNAs (miRNAs) are small non-coding RNAs that play an essential role in the regulation of biological processes during tumor progression [102]. In one study, miRNAs that target c-Src were screened, and miR-137 was identified as the critical regulator of Src in CRC cells. Expression of miR-137 was reported to be markedly downregulated in 100% (10/10) of patients with CRC, and the downregulated expression was associated with the early phase of cancer progression. miR-137 markedly suppresses tumor growth in vivo and in vitro, by targeting the 3′-untranslated region (UTR) of Src and AKT2 mRNAs and inhibiting their expression [103]. miR-424/miR-503 suppresses Rictor expression by binding to the 3′-UTR of Rictor mRNA. However, EGF-mediated Src activation results in the upregulation of mTOR2 and Rictor expression. Subsequently, this leads to Src-mediated downregulation of miR-424/miR-503, which suppresses invasive potential and tumor growth in vivo and in vitro [104]. miR-9 is a crucial regulator of hyperglycemia-mediated CRC development. miR-9 suppresses the high glucose concentration-induced proliferation, migration, and EMT of CRC cells, by inhibiting insulin-like growth factor 1 receptor (IGF1R)/Src activation and upregulating E-cadherin expression [105]. Moreover, the expression of miR-542-3p is downregulated in patients with CRC. miR-542-3p downregulates the expression of inhibiting integrin-linked kinase (ILK), which activates the integrin–FAK/c-Src pathway. miR-542-3p suppresses the pathogenesis of CRC by reducing the metastatic potential and tumor growth of CRC in vivo and in vitro [106]. Furthermore, miR-27b inhibits the invasion and growth of CRC tumors in vivo through inhibition of Src and FAK activation, by downregulating the expression of PXN, ARFGEF1, Rab14 and ADAM19, which are the miR-27b targets. However, the expression of miR-27b is downregulated upon the activation of the PI3K/AKT pathway [107]. The expression of miR-129-1-3p is markedly downregulated in CRC cells and patients with CRC. miR-129-1-3p suppresses the invasion and growth of CRC tumors in vivo and in vitro by binding to the 3′-UTR of Src, c-Yes, and Fer mRNAs, and inhibiting their expression [108].

### 5.4. Other Src Repressors

A CSK-homologous kinase (CHK), a serine/threonine kinase, inhibits SFK activation, without affecting phosphorylation, through non-catalytic mechanisms, which involve direct interaction with the SFK members, including Hck, Lyn, and Src [109]. The expression of CHK is reported to be markedly downregulated in 83.3% of patients with CRC. CHK inhibits Src activation, which markedly reduces the metastatic potential of CRC cells [110].

Downstream of tyrosine kinase-4 (Dok-4), a membrane-localized adapter protein, is an inhibitor of tyrosine kinase signaling in cancer. Dok-4 was reported to be a prominent substrate of Src, Fyn, Ret and JAK2 in CRC. Dok-4 promotes the phosphorylation of tyrosine at the pleckstrin homology (PH) domain. The phosphotyrosine-binding (PTB) domain of Dok-4 is localized to the cell membrane. Dok-4 inhibits the Fyn-mediated transcriptional activation of Elk-1 and c-Jun [111]. Reversion-induced LIM (Lin-1, Isl1, Mec3) (RIL), a repressor of tyrosine kinase signaling, was reported to be hypermethylated in patients with CRC. RIL induces the dephosphorylation of Src through the formation of a RIL/pY419 and c-Src/PTPL1 complex, which inhibits cell growth and colony formation of CRC cells [112,113]. RACK1, an intracellular receptor for activated PKC, inhibits cell growth by suppressing the Src-induced activation of AKT/Myc, and expression of cyclin D1, through the inhibition of Src by direct interaction [114,115,116]. Src and HGF induce endocytosis of E-cadherin, which leads to the E3 ligase Hakai-mediated degradation of E-cadherin. However, RACK1 blocks the disruption of cell–cell contacts by inhibiting Src and HGF-mediated endocytosis of E-cadherin [117]. Moreover, the loss of Runt domain transcription factor 3 (RUNX3), which acts as a tumor suppressor, blocks the apoptosis of tumors by downregulating the expression of p21, ARF and p53 [118]. Oxidative stress induces the activation of Src by upregulating histone deacetylase 1 (HDAC1). Src-mediated phosphorylation of RUNX3 leads to the translocation of RUNX3 from the nucleus to the cytoplasm, which promotes the degradation of RUNX3 [119].

## 6. Modulation of EMT by Src during CRC Progression

### 6.1. Correlation between Src and EMT in Colon Cancer

The induction of EMT in cancer cells promotes the transformation of epithelial cells into mesenchymal cells. This transition is closely associated with tumor malignancy and chemoresistance [120,121]. The activation of SFK is essential for the induction of EMT during the progression of CRC (Figure 3A). The upregulation of Src induces the phosphorylation of β-catenin/E-cadherin, which promotes the dissociation of β-catenin/E-cadherin and reduces cell–cell adhesion [122,123]. Additionally, Src decreases cell–cell adhesion by promoting the N-cadherin phosphorylation activities of Ras and FAK, which are the downstream targets of Src [123]. However, treatment with an Src kinase inhibitor inhibits tumor formation by the CRC cells in vivo and upregulates the expression of E-cadherin and α-catenin, which promotes strong cell–cell adhesion and morphological changes [124,125]. The activation of EMT enables the primary CRC cells to proliferate following extravasation, which promotes efficient metastasis of CRC cells [121]. However, Src activation enhances the permeability of endothelial cells [126,127]. Moreover, the activation of Src results in the activation of the PI3K/AKT pathway, which activates EMT and confers resistance to anoikis. Subsequently, this promotes apoptosis, by inhibiting adherent-independent growth through the loss of cell detachment to the ECM [128,129]. Additionally, Src induces transformation and resistance to anoikis in CRC cells by phosphorylating Tyr380 in caspase-8 [130].

ERK/MAP kinase and its direct substrate, myosin light chain kinase (MLCK), which are downstream targets of Src, are required for Src-mediated EMT induction through the formation of integrin adhesion complexes. The Src SH2 and SH3 domains promote the mesenchymal-like phenotype, by upregulating the expression of vimentin and MLC, and downregulating the expression of E-cadherin [131]. E-selectin, an adhesion receptor of endothelial cells, interacts with CRC cells and promotes extravasation of CRC cells by activating MLC through E-selectin-mediated ERK activation. E-selectin-mediated ERK activation leads to the dissociation of the E-cadherin/β-catenin complex, which is mediated by activated Src [132]. E-selectin-induced activation of DR3 triggers the activation of Src and, subsequently, the PI3K/AKT pathway. The metastatic CRC cells lead to the upregulation of the DR3 splice variant with no death domain, which cannot trigger apoptosis. This indicates that the CRC cells shift the functional role of DR3 toward survival [133].

c-Yes, a member of the SFK family, modulates adherent junctions through suppression of cell clustering by inhibiting the accumulation of E-cadherin. c-Yes suppresses apoptosis of CRC cells and increases tumor growth and liver metastases in vivo by promoting cell migration [134]. c-Yes activation promotes the growth and tumor formation of β-catenin-active CRC cell lines by promoting the expression of BIRC5 and BCL2L1, the transcriptional targets of the β-catenin/Yes-associated protein 1 (YAP1)/TBX5 complex. c-Yes activation induces the Tyr357 phosphorylation in YAP1, which is essential for β-catenin-dependent malignant processes. These processes are mediated by c-Yes-mediated regulation of the YAP1 activity of the complex, by promoting the formation of the β-catenin/YAP1/TBX5 complex [135].

### 6.2. Activators of Src-Mediated EMT

Several cellular components, such as Src activator, promote CRC progression by inducing EMT (Figure 3B).

#### 6.2.1. Caveolin (CAV-1)

The activation of FAK and Src induces cell adhesion by increasing extracellular pressure in CRC. Previous studies have demonstrated that Src activation is correlated with extracellular pressure and EMT. Increased extracellular pressure associated with CRC stimulates integrin-mediated cancer cell adhesion and cell growth through Src activation [136,137]. Additionally, the increased extracellular pressure enhances integrin-binding affinity, by activating β1-integrin at threonine 788/789, which is mediated by the interaction between CAV-1 and Src kinases [138,139]. CAV-1 is reported to be involved in EMT. The expression of CAV-1 is markedly upregulated in patients with stage IV CRC, when compared to that in the patients with stage I CRC. CAV-1 induces tumor formation and metastasis in vivo. However, N-myc downstream-regulated gene 1 (NDRG1), a tumor suppressor, inhibits the pathogenesis of CRC through CAV-1 degradation-mediated inhibition of EMT [140]. The Rho/ROCK signaling and Src-dependent CAV-1 tyrosine phosphorylation promote the metastatic potential and focal adhesion dynamics of CRC cells [141]. Moreover, the activation of AKT by Src-activated CAV-1 promotes the secretion and expression of Kallikrein 6 (KLK6). The expression of KLK6 is reported to be markedly upregulated in 90% of patients with CRC, and to be correlated with poor survival [142,143]. The upregulated KLK6 expression increases the survival of CRC cells in vivo. KLK6-induced Snail expression induces EMT by downregulating the expression of E-cadherin [144].

#### 6.2.2. SPROUTY2 (SPRY2)

SPRY2, a modulator of tyrosine kinase receptor signaling, was reported to be upregulated in 90% of patients with CRC. SPRY2 promotes tumor growth and metastasis by enhancing the activation of AKT and MAPK/ERK, through the upregulation of c-MET expression and inhibition of EGFR degradation [145,146]. The upregulated SPRY2/EGF/Src/AKT signaling induces EMT through Zab1, claudin (CLDN)-1-mediated suppression of epithelial adhesion and polarity proteins. SPRY2 induces EMT by upregulating the expression of Zab1, which is mediated by enhanced expression of the ETS1 transcription factor and decreased expression of miRNAs that target Zeb1, such as miR-200b, 200c and miR-150 [147].

#### 6.2.3. Twist-1/PDGFR

Twist-1/PDGFR-mediated Src activation promotes the EMT-mediated formation of invadopodia, which are extracellular matrix (ECM)-degrading actin-rich cellular structures, and matrix degradation. The formation of invadopodia, which is a hallmark of tumor cells, promotes tumor invasion and metastasis by transducing the signals from the tumor microenvironment [148,149,150]. The activated Src induces Tyr110 phosphorylation in the activator subunit NoxA1, which regulates the activities of Nox1 and Nox3. The activated NoxA1 interacts with Tks4 and Tks5, which are invadopodia components and Src adaptors, and induces the formation of invadopodia by inducing ROS generation [151,152].

#### 6.2.4. Integrin α5β1

Integrin α5β1 activates Src by forming a complex with Src, which subsequently results in FAK activation. Integrin α5β1 induces the formation of actin stress fibers and focal adhesions in CRC [153]. Integrin α5β1 suppresses E-cadherin-induced assembly of associated intercellular contacts by inhibiting calcium-mediated membrane localization of E-cadherin. Additionally, the integrin-mediated activation of Src enhances the assembly of adhesion complexes, and tumor growth of CRC [154,155]. The Src SH2 domain-dependent phosphorylation site (Tyr925) of FAK promotes the formation of integrin-induced focal adhesion, by increasing the protrusion motility of cells [156].

α-actinin is physically associated with integrin-mediated cell adhesion. The phosphorylation of tyrosine in α-actinin-1 markedly increases upon pressure-induced cell adhesion. The activated α-actinin-1 promotes the formation of α-actinin-1/Src/β1 integrin receptor complex and formation of α-actinin-1/Src/FAK complex and subsequently facilitates focal adhesion formation by phosphorylating Tyr397 and Tyr576 in FAK [157].

#### 6.2.5. Protein Tyrosine Phosphatase Receptor Type A (RPTPα)

RPTPα activates Src by interacting with the SH2 domain [158]. RPTPα induces the formation of contractility-mediated adhesion sites and stress fibers by enhancing the expression of Src downstream targets, such as MLCK, in focal adhesions [159,160]. miR-218 suppresses tumor growth of CRC by binding to the 3′-UTR of RPTPα mRNA and suppressing the expression of Src [161].

#### 6.2.6. CLDN1

The expression of CLDN1, a tight junction protein, is upregulated in patients with CRC. CLDN1 activates Src and PI3K by forming the CLDN1/Src complex, which increases tumor growth and liver metastasis in vivo by promoting EMT [162,163]. TNF-α-mediated upregulation of CLDN1 activates the Src/ERK1/2 signaling pathway, and promotes cell viability and migration, by promoting EMT [164].

#### 6.2.7. Inositol-Requiring Enzyme 1α (IRE1α)

A previous study demonstrated that the activation of IRE1α induces cell growth of prostate cancer by activating the c-MYC signaling pathway [165]. The upregulation of fibronectin by IRE1α-mediated XBP1 activation is involved in the progression of CRC. IRE1α promotes the invasion and liver metastases in vivo and in vitro through EMT induction, by activating the Src/FAK/GTPase signaling pathway. Treatment with the inhibitor of IRE1α blocks the migration of CRC cells [166].

### 6.3. Suppressors of Src-Mediated EMT

Several tumor suppressors involved in the progression of CRC inhibit EMT by inhibiting Src activity (Figure 3C). The suppressors of cytokine signaling (SOCS)-1 inhibit JAK/STAT3 pathways by inhibiting the phosphorylation of JAK, through direct binding with the substrate-binding groove of JAK [167]. The inhibition of ROS production by SOCS-1 suppresses the activation of both JAK/STAT3 and Src. This leads to the upregulation of nuclear factor erythroid 2-related factor 2 (*NRF2*) and thioredoxin 1 (Trx1), which is a regulator of antioxidant defense mechanism, inhibits the migration and survival of CRC cells by inhibiting EMT induction [168]. CLDN7 was reported to exert opposing effects when compared to CLDN1 in the progression of CRC. One study reported that the expression of CLDN7 was markedly downregulated in 100% (250/250) of patients with CRC. CLDN7 inhibits EMT by downregulating the transcription factors associated with EMT, such as Zeb1 and Snail, by suppressing CLDN1 expression. The inhibition of EMT by CLDN7 promotes tumor growth in vivo. CLDN7 is closely associated with improved survival of patients with CRC [169]. SAM and SH3 domain containing 1 (SASH1) is a tumor suppressor whose expression is downregulated in breast cancer and CRC [170]. The expression of SASH1 is inversely correlated with poor survival of patients with breast cancer [171]. The loss of SASH1 in the CRC cells promotes aggressiveness of CRC cells through Zab-1 and Zab-2-induced EMT. SASH1 suppresses the aggressiveness of CRC cells and poor survival of patients with CRC, through the inhibition of EMT by CRKL-mediated Src activation [172].

## 7. Src Induces and Maintains Cancer Stem Cells (CSCs) in CRC

Increasing populations of CSCs in cancer promote EMT, which results in the development of chemoresistance, induction of metastases and recurrence [121]. The cell surface glycoprotein, CD44, is one of the CSC-specific cell surface markers in various cancers, including CRC [121,173]. The upregulated expression of CD44 was reported to be involved in the pathogenesis of CRC. The expression of CD44 variants is markedly upregulated in patients with CRC, and is closely correlated with poor survival [174,175]. The CD44^+^ CRC cells exhibit markedly increased invasion, migration, and tumor formation in vivo when compared to the CD44^−^ CRC cells, which is mainly due to the downregulation of colonic epithelial markers (e.g., E-cadherin, CK20 and CDX1) and upregulation of EMT-associated transcriptional factors, including Snail, Twist and Slug.

The correlation between SFK and the acquisition of CSC traits has been identified in CRC. CD44 and its ligand, hyaluronan (HA), are involved in the maintenance of stem cell properties of CRC cells by upregulating Snail expression through Src-mediated inhibition of miR-203, a tumor suppressor. miR-203 inhibits proliferation and stemness of cancer [176]. CD44 promotes the proliferation and stemness through CD44–Src–integrin axis-mediated suppression of anoikis during sphere formation [177]. Moreover, CD44 activates Lyn, an SFK, through the formation of the CD44/Lyn complex, and subsequently enhances chemoresistance to carmustine through the activation of the PI3K/AKT pathway [178].

CD133 is a cell surface marker of colorectal CSCs. The activation of integrated CD133 (prominin-1)/integrin/Protease-activated receptor (PAR) signaling leads to the activation of Src/AKT/GSK3β signaling. Subsequently, this leads to the enhanced self-renewal ability of sphere-forming and side population (SP) cells through activation of the β-catenin-dependent transcriptional activity [179]. The expression of c-Yes is upregulated in the CD133^+^ subpopulation of CRC cells when compared to that in the CD133^−^ subpopulation. The loss of c-Yes in the CD133^+^ subpopulation induces cell differentiation, and decreases cell proliferation and tumorsphere formation, by suppressing Nanog expression [180].

Increasing the Cripto-1 (CR-1)-expressing subpopulation by activin/nodal signaling enhances tumor-forming ability by promoting self-renewal potential [181]. CR-1, which is detected in colon cancer spheroids from patients with CRC, is closely correlated with high levels of Nanog and Ephrin B2 receptors (EphB2). CR1 promotes the tumor growth of colorectal CSC/progenitors in vivo and in vitro, by activating the Glypican/Src/AKT pathway and inducing CD44 expression [182].

Recently, the correlation between Src and the MEK5/ERK5-mediated acquisition of CSC traits was identified in CRC. Src is identified as an upstream kinase of the MEKK3/MEK5/ERK5 pathway in YAP-mediated myogenic differentiation [183], and it regulates the generation of ROS via activation of ERK5 [184]. Further, c-Yes induces malignant progression of CRC by activating YAP1 [135]. NF-κB-induced IL-8 expression, by activation of MEK5/ERK5 signaling, is required for the acquisition of CSC properties and chemoresistance of CRC [185]. These results demonstrate that the Src kinase family may be involved in the acquisition of CSC traits of CRC, via activation of the YAP1/MEK5/ERK5 signaling pathway.

## 8. Therapeutic Implications of Src Inhibitors in CRC Progression

Aberrant Src kinase activity is reported to play an important role in CRC progression. Hence, there is a need to develop therapeutic strategies using Src as a drug target for various types of cancer, including CRC. 

### 8.1. Evaluation of Preclinical and Clinical Src Inhibitors

The Src inhibitor SU11333 markedly inhibits tumor growth, by inhibiting Src-mediated vascular permeability and Src-mediated activation of STAT3, CAV-1 and FAK [127]. PP2, an Src kinase inhibitor, suppresses the invasion of CRC cells by suppressing the Src-mediated upregulation of u-PAR expression. The treatment combination of PP2 and peptide NI-5.16, a u-PAR antagonist, effectively inhibits the invasion of CRC cells [186].

Dasatinib, an Src inhibitor, and oxaliplatin are approved by the Food and Drug Administration (FDA) for treating chronic myeloid leukemia (CML) (ClinicalTrials.gov identifier: NCT01460160) and colon cancer (ClinicalTrials.gov identifier: NCT00017082), respectively. Oxaliplatin, an inhibitor of DNA synthesis, effectively suppresses the expression of Src. Treatment with oxaliplatin markedly decreases the viability of CRC cells by downregulating the activation of c-Src and FAK. The treatment combination of dasatinib and oxaliplatin markedly suppresses cell proliferation and tumor growth in vivo and in vitro, through synergistic inhibition of VEGF expression and c-Src activation [187]. The treatment combination of intetumumab, an integrin inhibitor, and dasatinib effectively inhibits the migration and proliferation of dasatinib-resistant CRC cells, by suppressing the activation of FAK, PXN and Src [188].

Bosutinib (SKI-606), an Src inhibitor approved by the FDA for Philadelphia chromosome-positive leukemias (Ph+) (ClinicalTrials.gov Identifier: NCT00261846), is currently under evaluation for treating other cancers, including glioblastoma and lung cancer. Bosutinib inhibits the tumor growth of CRC in vivo, without affecting the bodyweight or causing cancer cell death, by inhibiting the phosphorylation of tyrosine in Src [189].

### 8.2. HDAC Inhibitors

The therapeutic effects of an HDAC inhibitor against CRC cells are correlated with the regulation of Src activation. Treatment with butyrate, an HDAC inhibitor, markedly inhibits Src activity and secretion of MMP2 and MMP9. Butyrate decreases cell proliferation, anchorage-independent growth, and expression of alkaline phosphatase, a colon epithelial differentiation enzyme [190,191]. Both butyrate and trichostatin A (TSA) have been approved for cancer therapy by the FDA. Butyrate downregulates the expression of Src, while TSA inhibits the transcription of SFKs, including Src, Lck, Lyn and Yes [192]. Moreover, TSA suppresses cell proliferation by downregulating Eps8 expression without affecting Src activation in the v-Src-transformed cells [193]. Garcinol, a polyisoprenylated benzophenone HDAC inhibitor, inhibits the interaction between Src and FAK or the p85 subunit of PI3K, by inhibiting the activation of Src. This decreases the proliferation and invasion of CRC cells, by inducing apoptosis through suppression of Src-mediated AKT and ERK activation [194,195]. The treatment combination of valproic acid (VPA), an HDAC inhibitor for epilepsy, and neratinib, a tyrosine kinase inhibitor, is currently under evaluation in phase 2 clinical trials for treating patients with solid tumors, such as colon and pancreatic tumors (ClinicalTrials.gov Identifier: NCT03919292). Treatment with VPA increases apoptosis in more than 50% of CRC cells by inhibiting AKT activation through the downregulation of Src expression. The treatment combination of VPA and bosutinib (an Src inhibitor) markedly inhibits the proliferation of CRC cells and tumor formation via xenografts [196].

### 8.3. MetAP2 Inhibitors

MetAP2 inhibitors can be useful for therapeutic drugs for the treatment of CRC. A-800141, the sulfonamide MetAP2 inhibitor, suppresses 70% of tumor growth by upregulating the expression of p21 and p53 and the formation of GAPDH variants with unprocessed N-terminal methionine [197]. Fumagillin and TNP-470, the MetAP2 inhibitors, directly inhibit the growth of 33–80% of colon, fibrosarcoma, prostate and lung cancer cells. MetAP2 inhibitors markedly decrease the anchorage-independent growth of cancer cells by more than 90%. The MetAP2 H231A deletion mutant, which lacks the catalytic activity, exhibits markedly increases sensitivity to MetAP2 inhibitors [68]. NC2231, a novel MetAP2 inhibitor, effectively inhibits cell proliferation with a half-maximal inhibitory concentration (IC_50_) value of 1.2 μM, by markedly inhibiting the phosphorylation of Src [69]. The clinical phase I trial of M8891, a MetAP2 inhibitor, is currently ongoing (ClinicalTrials.gov Identifier: NCT03138538).

### 8.4. Andrographolide

The therapeutic efficacy of andrographolide, a phytochemical from *Andrographis paniculata* (AP) plant, against CRC has been previously examined. Andrographolide inhibits v-Src activity by ubiquitination-mediated degradation of v-Src and its downstream ERK1/2 pathway. Andrographolide decreases the invasion ability of cells by suppressing the induction of EMT in the CRC cells [198]. The treatment combination of andrographolide and capecitabine is currently under evaluation for treating patients with locally advanced, recurrent, or metastatic CRC (ClinicalTrials.gov Identifier: NCT01993472) and patients with advanced or metastatic esophageal cancer (ClinicalTrials.gov Identifier: NCT04196075).

### 8.5. Trametinib

Pyrido[2,3-d]pyrimidines, which are highly specific inhibitors of Src kinase, effectively inhibit CRC cell proliferation by arresting the cell cycle at G2/M [199]. The treatment combination of trametinib, a pyrido-pyrimidine derivative, and dabrafenib, a BRAF inhibitor, is currently under evaluation in phase 2 clinical trials for treating patients with BRAF mutant resectable stage III melanoma (ClinicalTrials.gov identifier: NCT02858921).

### 8.6. PKC Inhibitors

Treatment with tyrosine kinase and PKC inhibitors, such as genistein and calphostin, inhibits pressure-induced CRC proliferation [200]. The microtubule-disrupting inhibitor colchicine inhibits extracellular pressure-mediated CRC cell adhesion by suppressing the activation of Src and FAK [201]. Colchicine has been approved by the FDA for the treatment of familial Mediterranean fever (FMF) and acute gout flares. Colchicine inhibits the formation of microtubules and disrupts microtubules [202]. Additionally, colchicine inhibits extracellular pressure-mediated tumor cell adhesion and peritoneal metastases, and markedly decreases the survival of the xenograft tumor by inhibiting the activation of Src, FAK and AKT [203]. Colchicine is currently under evaluation in phase 2 clinical trials for treating primary hepatic malignant tumors (ClinicalTrials.gov identifier: NCT04264260).

### 8.7. Curcumin

Curcumin is reported to exhibit anti-cancer effects. The treatment combination of neoplastic inhibitors and curcumin is currently under evaluation for treating various types of cancer, including CRC (ClinicalTrials.gov Identifier: NCT00973869, NCT03769766, NCT02439385). Curcumin enhances the anti-tumor activity of neoplastic inhibitors. The combination of curcumin and dasatinib exerts a synergistic inhibitory effect on the growth of CRC cells and intestinal adenomas of mice by inhibiting the activation of EGFR, IGF1R and c-Src [204].

### 8.8. Folic Acid or Folinic Acid

FDA has approved the treatment combination of panitumumab and Folfox for patients with CRC. Folic acid and folinic acid, which are forms of water-soluble vitamin B9, are currently under evaluation alone or in combination with other neoplastic drugs, including irinotecan, for treating patients with CRC (ClinicalTrials.gov Identifier: NCT02967289, NCT03142516). The folic acid treatment induces the activation of the folate receptor alpha (FRα)/SRC/ERK1/2/NFκB signaling pathway, which upregulates TP53 and suppresses cell proliferation and tumor growth in vivo and in vitro, by upregulating the levels of CDKN1A and CDKN1B [205].

### 8.9. Capecitabine

Capecitabine, a prodrug of 5-FU, is used as adjuvant chemotherapy for CRC. Capecitabine alone, or in combination with oxaliplatin, or CB-839, a glutaminase inhibitor, is currently under evaluation for treating CRC (ClinicalTrials.gov Identifier: NCT02978612, NCT04188158, NCT02861300). The treatment combination of capecitabine and lactate calcium salt (LCS), which is used to treat calcium deficiencies, markedly decreases the expression of Src and suppresses survival and tumor growth in vivo, when compared to the individual treatment [206].

## 9. Conclusions

The functions of SFKs in the pathogenesis of colon cancer have not been completely elucidated. However, tumor progression is regulated by SFKs through highly complex mechanisms. Thus, SFKs are one of the primary therapeutic targets for cancer, exhibiting increased survival, invasiveness and metastatic capacity, and resistance against neoplastic drug treatments and CSC properties. The development of kinase inhibitor drugs for targeting Src is essential for cancer therapy. Previous preclinical and clinical trials, using monotherapy or combination therapy with other neoplastic drugs, have revealed that the kinase inhibitors exert a synergistic effect for inhibiting tumor progression and prolonging the survival of patients with tumors, including CRC [3,207].

The elucidation of molecular mechanisms of SFKs in tumor progression may provide novel insights into the transition of tumor cells into metastasis static cells. Additionally, effective cancer therapy can be designed by identifying new cellular targets, which can be used in conjunction with therapeutic drugs against Src. The therapeutic drugs targeting Src can be used in combination with other kinase inhibitors or neoplastic inhibitors to inhibit cancer progression, by inhibiting multiple signaling pathways effectively.

## Figures and Tables

**Figure 1 cancers-12-01339-f001:**
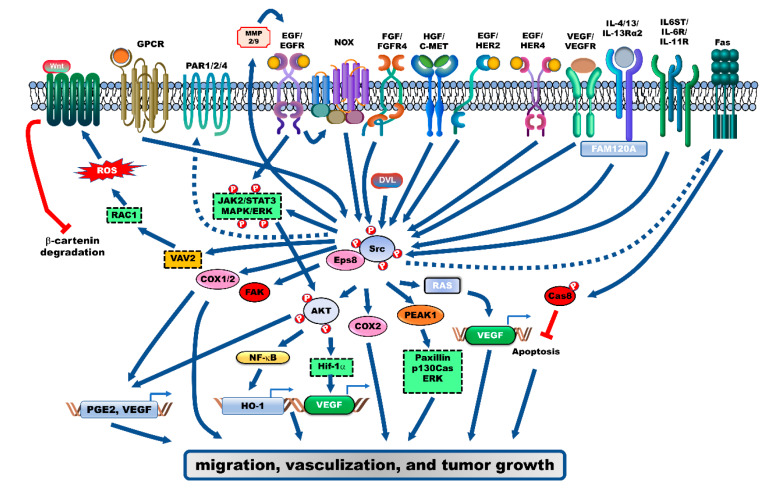
Receptor-mediated signaling pathways that activate Src during the progression of colorectal cancer (CRC). Several receptor-mediated signaling pathways activate Src, which plays an essential role in the progression of CRC. Src is activated by the ligand/receptor signaling complexes, including EGF/EGFR, HGF/c-MET, VEGF/VEGFR, FGFR, IL4/IL-13Rα2, and IL6/IL-11 signaling pathways, which further activate their downstream target signaling pathways, such as the AKT/NF-κB/HO-1, MAPK/ERK, and other oncoproteins to enhance proliferation, vascularization, and metastasis of CRC cells. Further, several G-coupled protein receptors (GPCRs) are involved in CRC progression through the activation of Src-mediated signaling pathways. PGE2/EP1, PARs, and CCK_2_R enhance cell proliferation by activating the EGFR/Src/MAPK/ERK and HIF-1α/Src/AKT/VEGF signaling axis. Additionally, the activation of Wnt-β-catenin signaling by Src-induced Rac1, which enhances reactive oxygen species (ROS) generation, results in enhanced migration of CRC cells.

**Figure 2 cancers-12-01339-f002:**
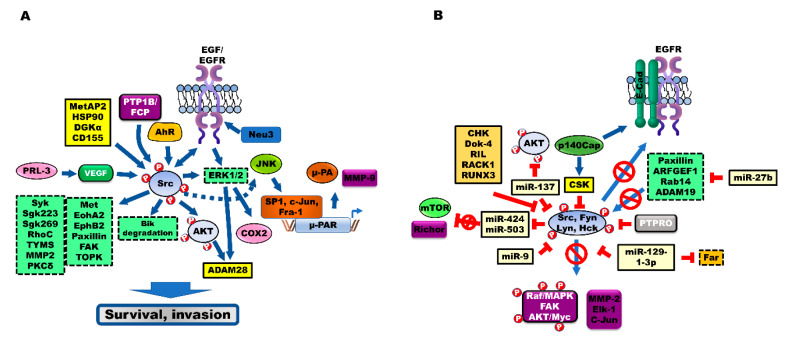
Modulators and suppressors were regulating Src activation during the progression of colorectal cancer (CRC). (**A**) Several cellular components positively promote the progression of CRC through Src activation. The expression levels of aryl hydrocarbon receptor (AhR), receptor-type protein tyrosine phosphatase (PTPR; PTP1B, RPTPα, and PRL-3), MetAP2, Neu, HSP90, DGKα and CD155 are upregulated in patients with CRC. These components activate Src and its downstream target signaling to enhance survival, invasion and chemoresistance of CRC cells. (**B**) Cell proliferation and metastasis of CRC cells are inhibited by inhibiting the activation of Src by multiple intracellular regulators, including C-terminal Src Kinase (CSK), protein tyrosine phosphatases (PTPRs), protein tyrosine phosphatase receptor type O (PTPRO), protein tyrosine phosphatase receptor type T (PTPRT), microRNAs (miRNAs), CSK-homologous kinase (CHK), downstream of tyrosine kinase-4 (Dok-4), reversion-induced LIM (Lin-1, Isl1, Mec3) (RIL), Receptor for activated C kinase 1 (RACK1), and RUNX family transcription factor 3 (RUNX3).

**Figure 3 cancers-12-01339-f003:**
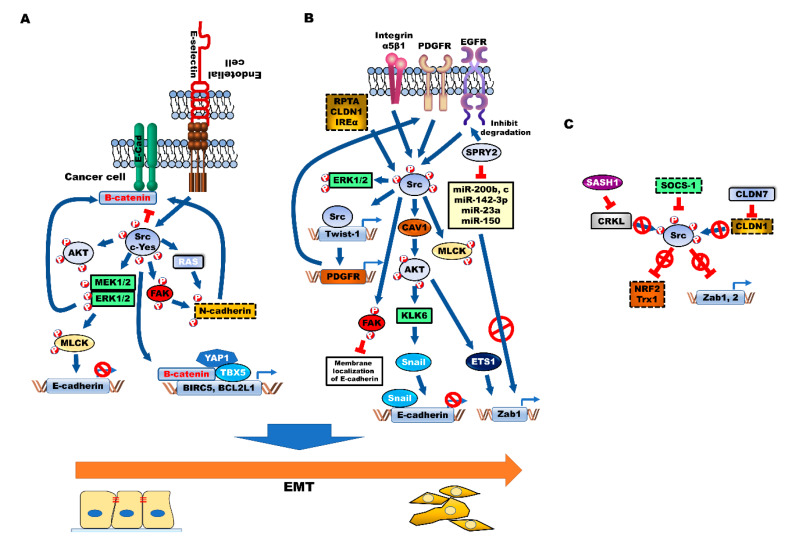
Src activation enhances aggressiveness, chemoresistance and acquisition of cancer stem cell (CSC) properties of colorectal cancer (CRC) by inducing epithelial-to-mesenchymal transition (EMT). Src family kinase (SFK) (**A**) and several cellular processes, such as Src activator (**B**)-mediated activation of AKT, MAPK/ERK and FAK, promote EMT by downregulating E-cadherin and inducing β-catenin-dependent malignant processes and N-cadherin expression, through induction of EMT-associated transcriptional factors, such as Twist-1, Snail and Zab1. (**C**) SOCS-1 inhibits EMT through the induction of *NRF2* and Trx1 by inhibiting Src. Additionally, CLDN7-mediated inhibition of CLDN1 and SASH1-mediated inhibition of CRKL suppress Src activation and inhibit EMT.

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
