# Peer review of "Regulation of Src Family Kinases during Colorectal Cancer Development and Its Clinical Implications"

_cancers, 2020, doi:10.3390/cancers12051339_

Round 1
Reviewer 1 Report
The present review gives a comprehensive and exaustive overview of SFKs, The present review gives a comprehensive overview of SFKs, their physiological role, and their implication in CRC progression. It has the merit to summarize the main signalling pathways involved in SFKs activation, the action of tumor suppressors and miRNAs as negative regulators of SFKs,together with the clinical aspects related to the use of SFKs as target molecules for personalized oncological therapies. Moreover, the figures of this review are well done, and present an overview at glance of the main aspects of Src patho-physiology. For all these reason I consider this review publishable in the present form.
Author Response
Comments and Suggestions for Authors (Reviewer #1)
The present review gives a comprehensive and exaustive overview of SFKs, The present review gives a comprehensive overview of SFKs, their physiological role, and their implication in CRC progression. It has the merit to summarize the main signalling pathways involved in SFKs activation, the action of tumor suppressors and miRNAs as negative regulators of SFKs, together with the clinical aspects related to the use of SFKs as target molecules for personalized oncological therapies. Moreover, the figures of this review are well done, and present an overview at glance of the main aspects of Src patho-physiology. For all these reason I consider this review publishable in the present form.
We appreciate your comments and the assessment of our review manuscript.
Reviewer 2 Report
In this review “Regulation of Src family kinases during colorectal cancer development and its clinical implications”, Wook J analyzed the role of Src family during colorectal cancer (CRC) development and its therapeutical involvement. The author provided an overview of the published studies and possible practical implications for CRC management. The manuscript is an accurate revision of the literature about the topic. However, some parts of the paper are difficult to read. A revision of the English style is highly recommended. A minor revision is needed before it is accepted.
1) Figure 1: the two figures should be aligned.
2) Line 131: the author should explain the specificity and the function of regorafenib.
3) Line 405: sentence should be corrected.
4) There is evidence that Src is linked to ERK5 signaling. This should be considered by the author. In particular, this article should be added and commented in the chapter 7: Diane M. Pereira, Sofia. E. Gomes, Pedro M. Borralho & Cecília M. P. Rodrigues MEK5/ERK5 activation regulates colon cancer stem-like cell properties Cell Death Discovery volume 5, Article number: 68 (2019).
5) In the References, lines 737, 822, 838, 905, 1135 should be checked.
Author Response
Comments and Suggestions for Authors (Reviewer #2)
In this review “Regulation of Src family kinases during colorectal cancer development and its clinical implications”, Wook J analyzed the role of Src family during colorectal cancer (CRC) development and its therapeutical involvement. The author provided an overview of the published studies and possible practical implications for CRC management. The manuscript is an accurate revision of the literature about the topic. However, some parts of the paper are difficult to read. A revision of the English style is highly recommended. A minor revision is needed before it is accepted.
1) Figure 1: the two figures should be aligned.
As your suggestion, we combined Figure 1A and Figure 1B into a single Figure 1 and aligned.
2) Line 131: the author should explain the specificity and the function of regorafenib.
As your suggestion, we added the paragraphs (line 140) to explain the specificity and the function of regorafenib as follows. Regorafenib is a sorafenib-derived chemotherapy drug as a multikinase inhibitor, which blocks the activity of (VEGFR1–3), PDGFRβ, BRAF, KIT, RET, angiopoietin 1 receptor (TIE2), and FGFR1 [34].
3) Line 405: sentence should be corrected.
We apologize for this error and have corrected this as follows (Line 429). The metastatic CRC cells lead to the upregulation of the DR3 splice variant with no death domain, which cannot trigger apoptosis.
4) There is evidence that Src is linked to ERK5 signaling. This should be considered by the author. In particular, this article should be added and commented in the chapter 7: Diane M. Pereira, Sofia. E. Gomes, Pedro M. Borralho & Cecília M. P. Rodrigues MEK5/ERK5 activation regulates colon cancer stem-like cell properties Cell Death Discovery volume 5, Article number: 68 (2019).
We added and commented on the contents of Src-linked ERK5 signaling to chapter 7 as follows, as you recommended.
Recently, the correlation between Src and the MEK5/ERK5-mediated acquisition of CSC traits was identified in CRC. Src is identified as an upstream kinase of the MEKK3/MEK5/ERK5 pathway in YAP-mediated myogenic differentiation [184] and regulates the generation of ROS via activation of ERK5 [185]. Also, c-Yes induces malignant progression of CRC by activating YAP1 [136]. NF-kB-induced IL-8 expression by activation of MEK5/ERK5 signaling required for the acquisition of CSC properties and chemoresistance of CRC [186]. These results demonstrate that the Src kinase family may involve in the acquisition of CSC traits of CRC by activation of the YAP1/MEK5/ERK5 signaling pathway.
5) In the References, lines 737, 822, 838, 905, 1135 should be checked.
We apologize for these errors and have corrected. Please see lines 812, 919, 935, 1018, 1305.
Reviewer 3 Report
The author review the relationship between SFK and CRC. Although I am not a very expert on this very specific field, as far as I know, there are no competitive reviews that specifically focus on the relationship between SFK and CRC. It reviews more than 200 references, add many intuitive diagram that helps readers' understanding very much, the reference list includes enough number of new references. Thus, I did not have any specific objections to the publication of this manuscript as it is.
Author Response
Comments and Suggestions for Authors (Reviewer #3)
The author review the relationship between SFK and CRC. Although I am not a very expert on this very specific field, as far as I know, there are no competitive reviews that specifically focus on the relationship between SFK and CRC. It reviews more than 200 references, add many intuitive diagram that helps readers' understanding very much, the reference list includes enough number of new references. Thus, I did not have any specific objections to the publication of this manuscript as it is.
We appreciate your comments and the assessment of our review manuscript.